# A low-cost markerless motion capture system to automate functional gait assessment: Feasibility study

Osman Darici[1], Chanel Cabak[1], Jeremy D. Wong[1,2]*

1 Faculty of Kinesiology, University of Calgary, Calgary, Alberta, Canada, 2 Department of Biomedical Engineering, University of Calgary, Calgary, Alberta, Canada

* jeremy.wong2@ucalgary.ca

## Abstract

Functional gait assessments in older adults have traditionally required manual in-person quantification of clinical measures such as walking speed and step placement. This reliance on individuals trained in motion analysis hinders the frequency with which they are performed and reduces their generalizability and replicability. To standardize, simplify, and broaden access to gait assessments we here deploy recently-developed open-source tools to produce a low-cost, AI driven markerless motion capture system with custom analysis software for Functional Gait Assessment. Our system uses 3 Cameras and was validated with a traditional marker-based system on subjects (N = 3), showing strong correlation to laboratory-standard measures of step length ($R^2 = 0.98$), step width ($R^2 = 0.97$), and head speed ($R^2 = 0.95$). The markerless system's FGA reports demonstrated data similar to previous FGA findings in older adult subjects (N = 5). Moreover, supplemental standard biomechanical gait measures Step Width, walking Duration, and continuous Gait Speed may be integrated to augment existing FGA. This study demonstrates a proof-of-principle open-source markerless system for analyses of functional gait.

## Introduction

Due to cost and complexity of biomechanical analysis tools such as motion capture, standard clinical human gait tests are typically designed to be easy to perform by a clinician. However, these tests still cost time and attention to administer and may also be sensitive to inter-clinician variability. This is true for the tests such as Functional Gait Assessment (FGA) [1], Timed Up and Go [2], and Edinburgh Visual Gait Score [3], each of which are different protocols meant to monitor gait for clinical decision making and fall risk prediction. Therefore, in this study we focused on implementing a low cost, user friendly, open-source AI driven motion capture system to automate spatiotemporal gait measurements used in FGA.

**Data availability statement:** All relevant data are available on Dryad at the following DOI: 10.5061/dryad.2bvq83c2t.

**Funding:** This research was supported by NSERC Discovery (2022-04459; Wong), the University of Calgary Research Office Catalyst Grant (1061438; Wong), and Biomedical Engineering Summer Student Scholarship (Cabak).

**Competing interests:** The authors have declared that no competing interests exist.

The Functional Gait Assessment does not require sophisticated equipment or analysis, but instead requires a sophisticated observer. Ten different gait conditions—such as walking with changes in speed, motion of the head, or while stepping over obstacles—are performed during which task time is recorded via stopwatch and lateral step placements, gait speed changes, response time, and step height over obstacles are visually monitored. Some conditions require subjective judgments such as whether a speed change was "significant" or "minor". Advances in markerless motion tracking leveraging artificial intelligence (AI) may enable low-cost motion capture systems to be utilized to perform such clinical test and enhance its limitations.

An obstacle to motion capture-based gait analysis research is the complexity and cost in analyzing walking behaviour. Motion capture data—the most widely accepted method for analysis of full-body locomotion—requires costly equipment to collect, substantial time to analyze, and skill to convert kinematic data into clinically relevant gait metrics. Dedicated spaces are often used to allow a set of cameras to encircle a subject. Markers placed on limb segments consume substantial time, and cameras must be correctly calibrated to convert 2D images recorded from cameras into 3D positions. Commercial markerless laboratory systems have begun to be used which eliminate the use of markers at comparable accuracy [4,5], but remain accessible only to research institutions due to both the operational costs of annual software licensing and complex analysis.

Advances in AI-driven computer vision have produced open-source pose estimation and markerless motion capture techniques that have lowered the barriers to motion capture [6–10]. However, the application to automated clinical gait assessments remains limited, in part due to non-standardization of clinical tests. Multi-camera open-source systems provide rich data, but frequently lack comprehensive end-to-end integration with established protocols to provide repeatable clinically-relevant metrics [11]. Systems utilizing monocular video are simpler to use and reliable at providing gait asymmetry estimates in stroke patients [12,13] but do not demonstrate highly-accurate estimates of step width, believed to be a critical indicator of balance control and fall risk control [14–19]. Therefore, a significant need exists to make markerless motion capture a consistently-used clinical tool.

A low cost, user-friendly, markerless motion system could give clinicians access to gait measures used in biomechanics literature that were previously too difficult, such as step width and continuous gait speed. For example, mediolateral foot placement is used in FGA, possibly because it is easier to observe manually, but confounds gait heading and step width together. Instead, step width is widely understood to reflect a human's mediolateral balance control as noted above [14–19]. Additionally, step width is known to change during tasks with greater balance demands ("gait challenges") such as eyes-closed [20]. Besides step width, walking speed is another critical parameter [21]. It is typically measured in a clinical setting as an average (distance divided total movement duration) and changes in speed is visually observed within a task. Yet continuous measurement of a gait speed trajectory reflects the control of non-steady gait [22] and thus applicable to FGA tasks, and which may provide information about an individual's value placed on energetic cost and movement time

[23]. Outcomes such as step width per step and gait speed trajectory can be made accessible with a markerless motion system together with automated gait analysis.

In this study, we chose The FreeMoCap Project (Matthis et al., 2022; release 1.1; https://github.com/freemocap/freemocap/releases) to measure body segment position, supplemented with custom code for FGA analysis (https://bitbucket.org/jdwongmcl/plosone2026daricietal/src/main/) in an effort to determine if such a system could reduce measurement subjectivity and automate assessment. FreeMoCap represents one of the first among a set of open-source 3D computer vision projects capable of integrating pose estimations from multiple cameras into estimated of segment locations with consumer-grade cameras. FGA is an assessment widely used in clinics and which is a versatile test involving complex movement patterns, such as turning, speed changes, and obstacle navigation. We utilized 3 consumer-grade web-cameras and compared the accuracy of the system to a standard marker-based motion capture system. We then measured older adults performing FGA and developed custom software to automatically output performance metrics of FGA without any human interaction. Furthermore, we extracted clinically relevant step width and speed trajectories for each condition of FGA which cannot be determined by visual inspection. We therefore tested the prediction that an automated system could reliably perform FGA, automatically output test results and improve clinical relevance by automating scoring based on step length, step width and walking speed.

## Methods

### Ethics

This study was approved by the Conjoint Health Research Ethics Board of the University of Calgary (REB#:21-1497v4). Written informed consent was obtained from all participants prior to enrollment. Subjects were recruited and tested from July 1 2024 to September 1 2024. The individual in this manuscript (Fig 1) has given written informed consent (as outlined in PLOS consent form) to publish these case details.

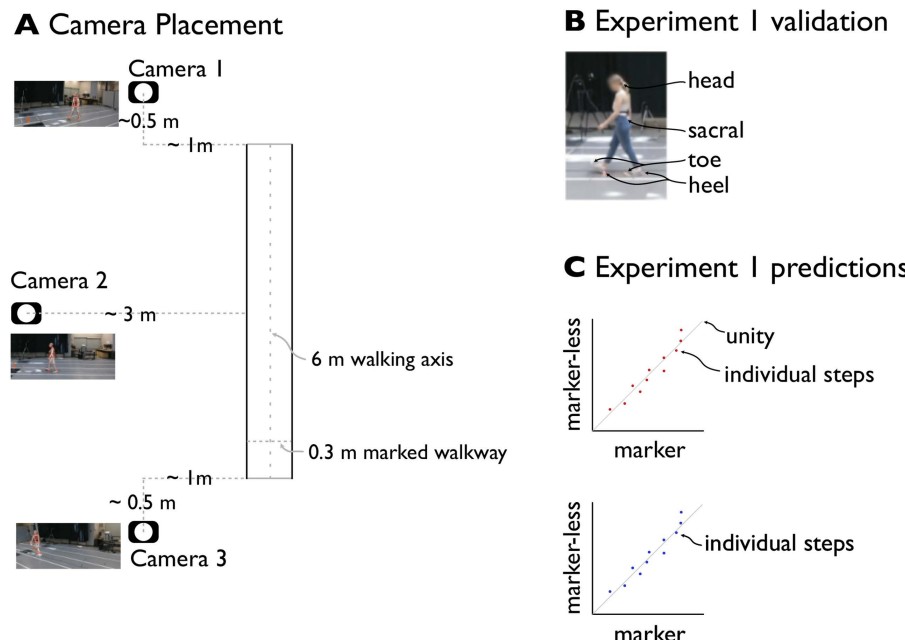

**Fig 1. Experimental setup and validation experiment.** A: Camera placement, with approximate distances relative to the left edge of the walkway. B: Experimental validation of step placements, using markers placed on the feet, sacrum and head. C: Validation of markerless data: step length, width and walking speed.

## Experimental designs

We performed two experiments. First we established the validity of the opensource markerless motion capture system (FreeMoCap: Matthis et al., 2022) to record clinically-relevant metrics—step width, step length, and walking speed—by comparing markerless foot placement and the resulting step times and durations, to those from an active optical motion capture system commonly used in biomechanical studies (Phasespace Inc.). Next, we performed Functional Gait Assessment on Older adults [1]. We placed the analysis code in a repository for use by practitioners interested in performing their own analysis.

## Methods

The markerless motion capture system used 3 (Jetaku Inc; Amazon) web cameras placed laterally to the walking direction along a 6 meters distance which is used in FGA (Fig 1A). We also marked the hallway to denote a 6 m walkway, with a width of 0.3 m. A desktop PC (2018; 3.4 GHz; 4 Cores; 16 GB Ram) was used to record and process data. Total added cost of the cameras and USB extension cables was less than $200 (CAD).

Experiment sessions required approximately 20 minutes of preparation time to 1) physically set-up cameras, 2) perform calibration and 3) compute capture volume orientation. To calibrate the system, the markerless motion capture system optimally estimates the 3D position of web cameras, and accounts for visual distortion of the camera optics. This is done by simultaneously recording video data from a Charuco board—a single black-and-white calibration image with known checkerboard pattern and dimension. To accommodate a 6m walking area that FGA requires, a large poster (square size: 15.7 cm) was used. Custom software optimizes the 3D position of the cameras to minimize the 3D-positional error of the detected checkerboard patterns (or "bundle adjustment [25]"). The software [24] reports the number of full boards detected from each camera, and provides some information about calibration accuracy; practically, we ensured that at least 400 boards were recorded for from each of the web cameras for each of our recording sessions. This calibration process took 2 minutes of Charuco board positioning, and about 10 minutes of compute (no GPU acceleration; CPU processing only on our 2018 machine). To perform capture volume orientation, subjects stood in three locations to define a simple coordinate system: 1) at the beginning of the walkway; 2) at +5m (forward) along the pathway; and 3) at 1 m right of the walkway. This procedure defined the plane of the walking area, allowing rotation of the recorded data. It also allowed us to perform linear correction for calibration bias, as these known magnitudes allowed rescaling data collected along each axis. Typically, FreeMoCap's calibration procedure resulted in less than 5% error in absolute estimate of walkway length.

During data processing, key-point software makes estimates of the position of joints and other points of interest on the body (Google Mediapipe). The software performs a lowpass filter (25 Hz) to reduce the effect of joint misestimation, motion blur, and other artefacts.

## Experiment 1: Validation

We performed our first experiment to determine the validity of the markerless motion capture system to compute three measures that are necessary and sufficient to automate FGA: step length, step width, and walking speed. To determine the accuracy of computer vision methods, we used marker-based motion capture and collected 6 markers per person to capture feet, torso, and head position of subjects (N = 3). While the markerless system we used reports several foot keypoints, throughout this study we will use the term "foot position" and "foot speed" to refer to Mediapipe's detected "heel position" marker and its derivative. We simultaneously recorded walking trials of a 4 m distance with both systems, under two conditions chosen to span the range of empirically-observed step widths: normal preferred step width, and a 'wide' condition where subjects were asked to place footsteps along the edges of the 0.3 m marked path. We performed three replications of each condition.

We time-aligned the two systems by determining the maximum cross-correlation (MATLAB: normxcorr2) of anterior-posterior (AP) foot speed between the two extracted walking bouts (Fig 2A). For each trial, we cross-correlated (MATLAB: normxcorr2), the summation of each system's (markerless and markered) left and right foot speed (Fig 2B) and computed the maximum value of the cross correlation. This cross-correlation was done using linearly-interpolated data from the markerless system data (recorded at 30 Hz) to match the markered system's sample rate (240 Hz). To spatially align the two systems, we subtracted the initial (T = 0) position of each left foot to register the two systems at the beginning of the walking bout.

To detect left and right stance periods, Foot speed was low-pass filtered at 3 Hz, and regions where foot speed was below the speed threshold of 0.15 m/s were identified. The average position of that stance region (the midstance instant,

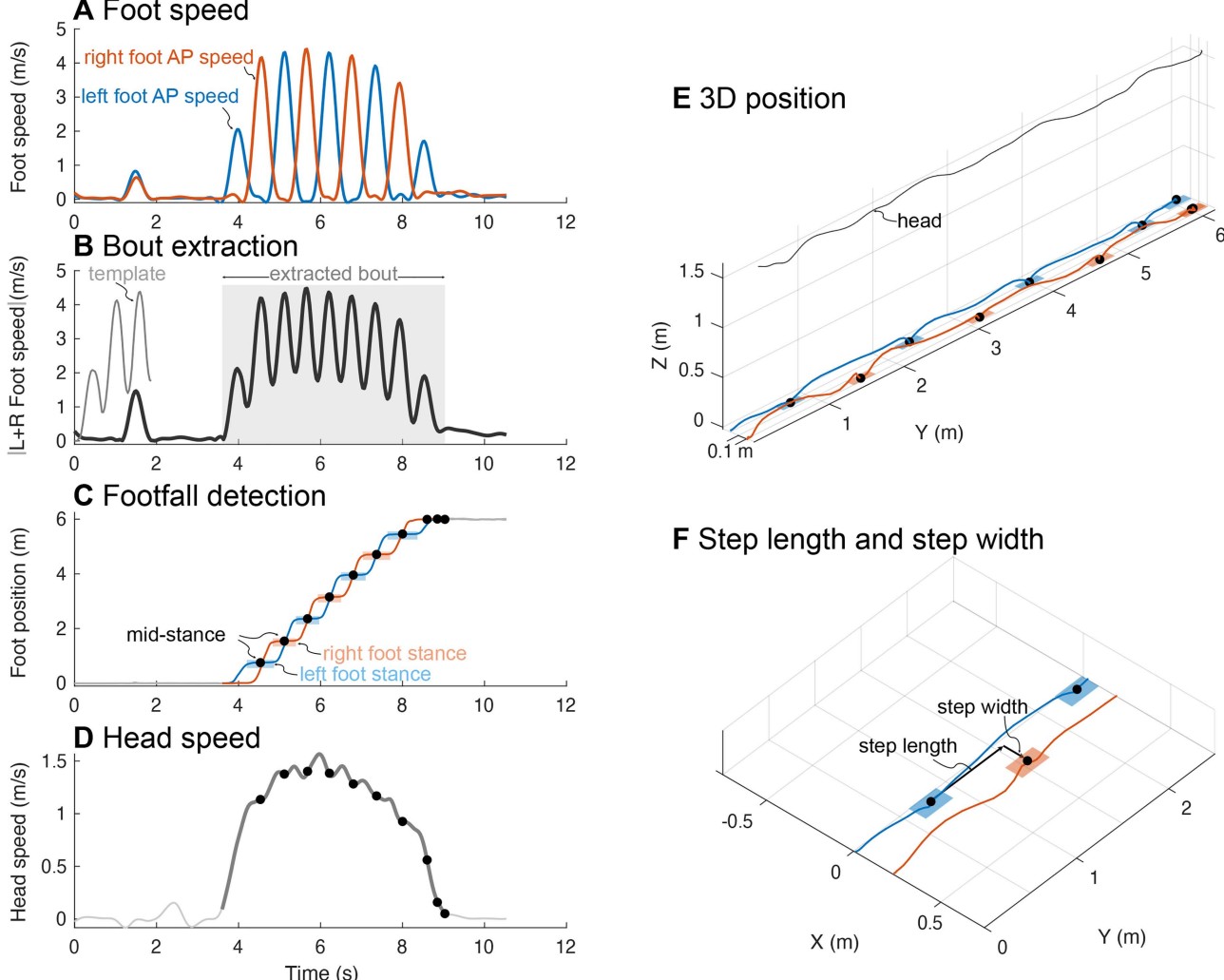

**Fig 2. Measurement of gait parameters. (A)** Anterior-posterior (AP) left and right foot speed was used to identify walking bouts automatically. **(B)** Cross-correlation of the summed AP absolute foot speeds with a starting-bout template were used to extract gait start and end times. **(C)** The regions where foot speed was below a threshold of 0.15 m/s were identified as stance phase, and the middle of each step's stance phase was taken as mid-stance. **(D)** Head speed was computed as the time derivative of the average position of the ear and nose markers locations. (E and F) step length and width vectors were computed based on the mid-stance instants of consecutive steps.

Carlisle and Kuo, 2023) was taken as the footstep location (Fig 2C–2F). We computed step length and width as the distance between two consecutive footstep locations in forward and lateral directions respectively (Fig 2F).

We measured head speed (Fig 2D) as differentiating (time derivative) the forward position of the head marker (located centrally in the back of the head via headband), and for the markerless system we defined it as the time derivative of the average forward position of the left ear, right ear, and nose keypoints.

### Experiment 2: FGA

Older adult subjects (N = 5; mean age: 69 years; Table 1) were asked to perform a complete functional gait assessment. Subjects completed three repetitions of each task. All older adults were living independently at the time of the experiment and completed the experiment only after informed consent stating that they felt comfortable performing the test of tasks.

### Markerless setup: Lighting

Accurate detection of body segments is sensitive to visual elements that make it difficult for the Mediapipe algorithm to clearly separate body keypoints from the background. Moreover, illumination—which affects shutter speed and thus motion blur—can affect localization. Optimal algorithm performance was achieved by using contrasting shoes with the ground colour—either white or dark shoes against the grey floor—and similarly contrasting colours for clothing, that fit relatively tight to the body. These adjustments can substantially affect the algorithm's tracking accuracy. For each subject we ensured that body keypoints during a practice trial were consistently and smoothly detected before proceeding with data collection and swapped out poorly-detected shoes or clothing if the keypoint detection was problematic.

### Data analysis

We developed custom MATLAB code to analyze walking bouts for Functional Gait Assessment, which uses a detailed set of manually-extracted gait features but relies on step placement and walking speed. To determine a walking bout's start and end instant, we used a template cross-correlation of summed foot speed similar to Experiment 1. For each trial, we cross-correlated (MATLAB: normxcorr2) the summation of the trial's left and right foot speed with a gait-initiation template and defined the walking bout's start time as the maximum of this cross-correlation; a gait-termination template was similarly used to detect the bout's end time (Fig 2B).

### Functional gait assessment scores: Automation

The Functional Gait Assessment (FGA) [1] is designed to evaluate an individual's ability to maintain stability while walking under a range of conditions. It consists of 10 walking tasks—such as changes in gait speed, walking with head turns, and stepping over obstacles while walking on a 30.48-cm marked walkway. Each condition is scored from 3 (normal performance) to 0 (severe impairment). For several conditions, scoring relies on qualitative judgments made by an observer, rendering test outcomes susceptible to observer bias. For example, the baseline condition "Gait Level Surface" requires the clinician to continuously monitor the maximum lateral foot placement deviation outside of the 30.48-cm wide walkway

**Table 1. Older adult subject demographic variables.**

| Subject | Height (m) | Weight (kg) | Age | Sex |
|---|---|---|---|---|
| 1 | 1.82 | 111 | 65 | Male |
| 2 | 1.58 | 69 | 79 | Female |
| 3 | 1.72 | 84 | 69 | Female |
| 4 | 1.58 | 68 | 67 | Female |
| 5 | 1.64 | 57 | 65 | Female |

while determining speed to be "good", "slower", or shows evidence of "imbalance" (no quantification for each adjective). In Condition 2 (Change in Gait Speed), the clinician must determine whether a speed change is "significant" relative to the "Gait Level Surface" trial (no quantitative definition) while again monitoring maximum lateral foot placement. In Conditions 3 and 4, which involve walking with horizontal and vertical head turns, the clinician must observe the subject to take three straight steps, followed by three steps with the head turned to the right and then three steps with the head turned to the left, on top of continuously monitoring for significant speed changes and maximum lateral foot displacement. These requirements highlight that FGA administration involves complex, simultaneous judgments without precise measurements, making scoring highly dependent on observer interpretation.

Using the outputs of our system, we automated the FGA scoring by setting thresholds for FGA categories on gait speed, relative to "Gait Level Surface" walking. Changes in gait speed of 0.10 to 0.20 m/s may be important across multiple patient groups [26], and based on the measured average speed of FGA condition 1 of our experiment, a 0.10 to 0.20 m/s change in gait speed correspond to approximately %5-%10 change. This formed the basis of our threshold approach and we have used 75% to 125% change in gait speed for FGA scoring where 100% corresponds to the speed of the condition 1. Note however that these threshold categorizations are meant to demonstrate the automation approach and are parameters in the algorithm that may be edited to suit the needs of the clinician.

Condition 2: *Gait Speed Change*:

- FGA score 3: speed change greater than 125% Condition 1 (normal walking)

- FGA score 2: speed change between [110%, 125%] Condition 1

- FGA score 1: speed change between [105%, 110%] Condition 1

- FGA score 0: no speed increase greater than 105% of Condition 1

Conditions 3 and 4*: Gait with Horizontal/Vertical head turns*:

- FGA score 3: maximum speed within [95,105]% of Condition 1, mediolateral placements (outside of the 30.48-cm wide walkway) as noted in [1] Condition 1

- FGA score 2: speed within [90,110] % of Condition 1, mediolateral foot placements as noted in [1]

- FGA score 1: speed change within [75, 90]% of Condition 1

- FGA score 0: speed change less than [75%] Condition 1

Condition 5: *Gait And Pivot Turn*:

- FGA score 3: turning in less than 3 s

- FGA score 2: turning (cued verbally at 2m into walk) took longer than 3 s

- FGA score 1: turning (cued verbally at 2m into walk) took longer than 4.5 s

- FGA score 0: incomplete

Condition 6: *Step Over Obstacle:*

- FGA score 3: high obstacle clearance, speed greater than 75% Condition 1

- FGA score 2: lower obstacle, speed greater than 75% of Condition 1

- FGA score 1: lower obstacle, speed less than 75% of Condition 1

- FGA score 0: incomplete

## Results

### Experiment 1

The markerless motion capture system was accurate in its detection of foot placement events (Fig 3A). The Markerless foot positions showed good agreement with the marker-based motion capture during both normal and wide step widths, demonstrating high correlation in step length (Fig 3B) and step width for both normal and wide trials ($R^2 = 0.97$; Fig 3C). Linear regression of Markered against Markerless Step Length yielded a slope of 0.995 (95% CI: 0.978–1.012) and intercept of 0.006 m (95% CI: −0.004–0.015 m), and for Step Width a slope of 0.972 (95% CI: 0.9535, 0.9906) and intercept −0.0031 (95% CI: −0.0082–0.0019), suggesting no significant bias. Agreement between Markered and Markerless was also assessed using Bland-Altman (Fig 3D–3E), showing a mean (markerless-markered) difference of −0.004 m in Step Length, and the 95% limits of agreement ranged from −0.063 m to 0.056 m. In Step Width, the mean difference was 0.003 m, and 95% limits of agreement ranged from −0.060 to 0.066 m.

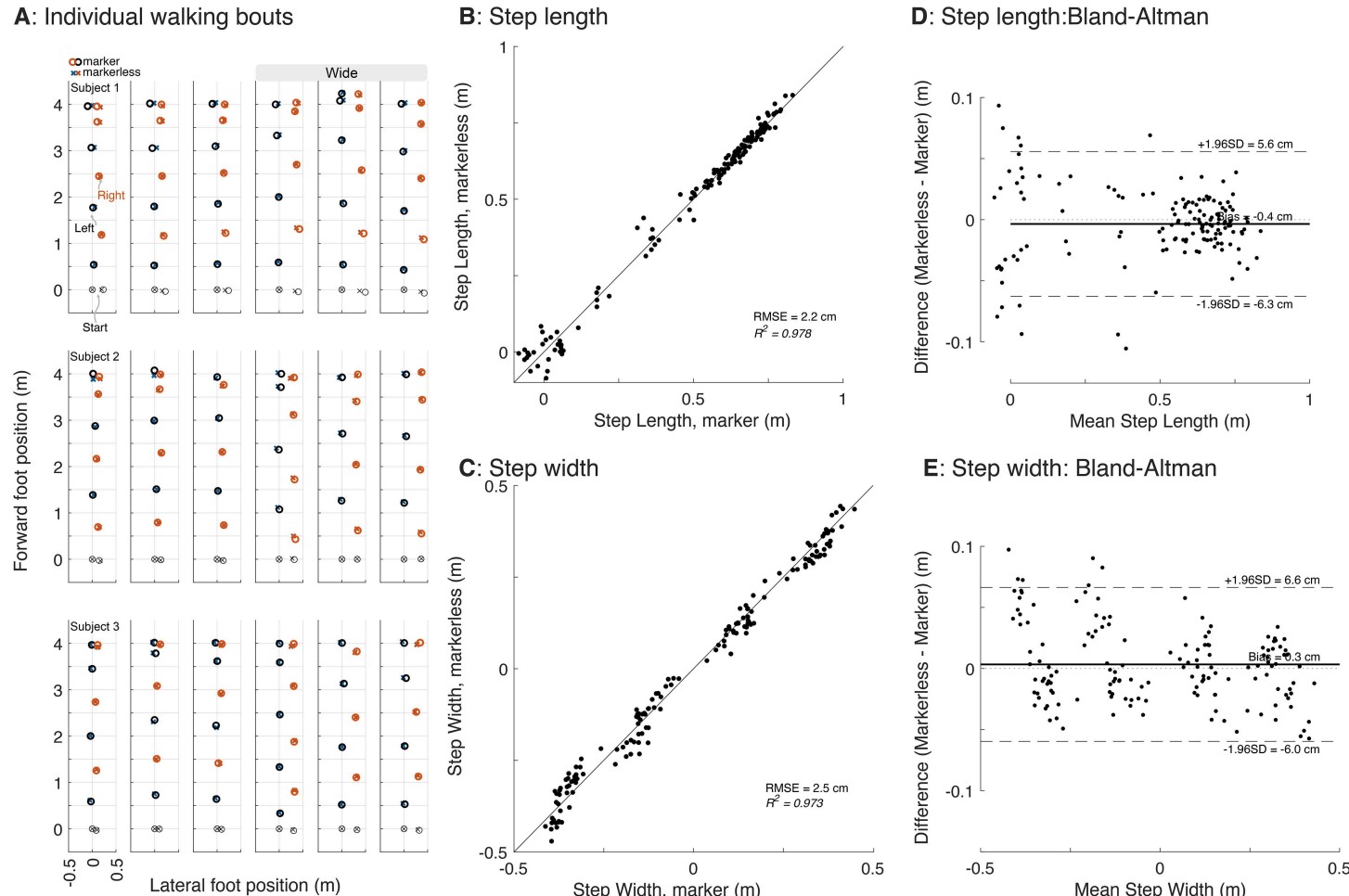

**Fig 3. Comparison of markerless system foot placement estimates with traditional marker-based motion capture.** A: individual steps of all subjects of both normal and wide walking trials identified via marker (O's) and markerless motion capture (X's). B and C: Step length (B) and width (C; negative is left-to-right step width) for each step shown in **A**. Bland-Altman plots depict mean-averaged differences between the two systems **(D, E)**.

Head speed measures with the markerless system were also similar to marker-based motion capture. Fig 4 depicts step speed across the normal and wide trials (Fig 3; $R^2 = 0.91$ to 0.98; mean = 0.95). Again, linear regression of markered against markerless showed very good agreement, with intercept 0.0394 m/s (95% CI: −0.0312 m/s – 0.0475 m/s) and Slope of 0.9741 (95% CI: 0.9655, 0.9828).

## Experiment 2

In Experiment 2, we employed the Functional Gait Assessment (FGA) for our older adult subjects, which resulted in quantification of individual footstep locations (position of the foot at mid stance instances, Fig 5A, top) for each condition. Variability was qualitatively evident in footstep locations especially in eyes closed and backwards walking. Head speed trajectories (Fig 5A, bottom) were sufficient to show deviations in walking speed (forward speed) per condition, relative to nominal preferred walking (Trial 1).

Average FGA scores (Fig 5B, top) were similar to previous literature values for older adult cohorts of age 70–79 and 80–89 [27] and most subjects scored lower for backwards, and eyes closed walking compared to other conditions. We also observed fluctuations in maximum lateral step placement (Fig 5B, bottom) comparing it to the horizontal dotted lines (Fig 5B and also 5C) that indicate the limit values of lateral foot placement for FGA scoring such as 15.24 cm outside of the 30.48-cm walkway. For the tasks Normal, Speed Change, Vertical Head Turns and Horizontal Head Turns, subjects exhibited qualitatively similar lateral displacement. Compared to Normal, both Eyes Closed and Backwards walking demonstrated substantially larger displacements that were also more variable within subjects. Supplementary measures of maximum and average step width and duration (Fig 5C) showed similar trends.

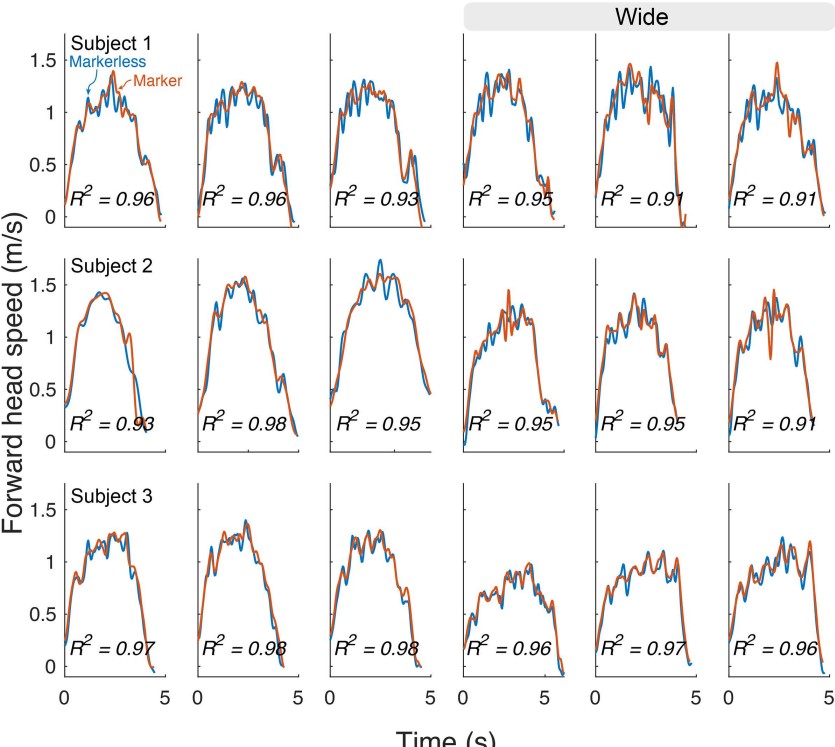

**Fig 4. Head speed measured during walking trials, by markerless (blue) and marker-based (red) methods.**

**A** FGA: Subject data

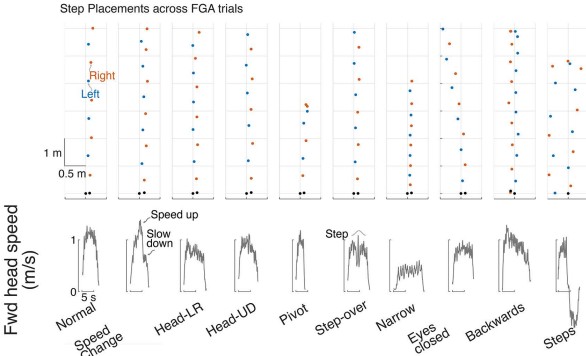

**B** FGA: Scoring

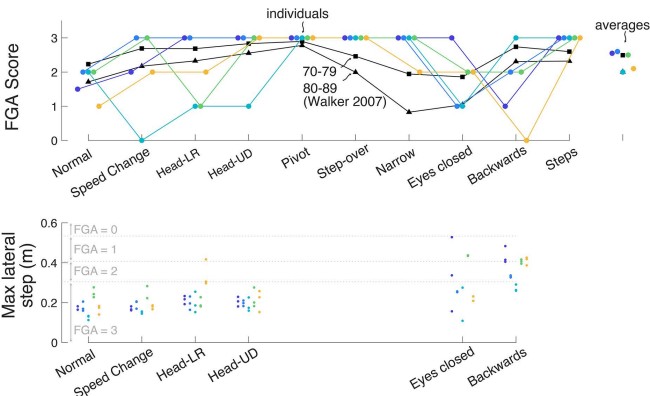

**C** Proposed supplementary measurements

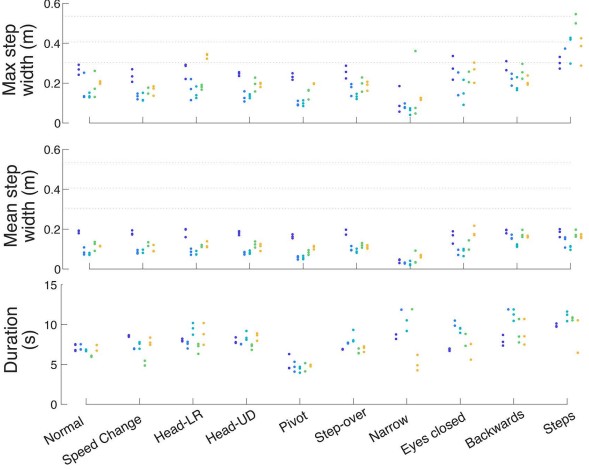

**Fig 5. Functional Gait Assessment.** A: Representative subject data, step placements for left (blue) and right (red) feet (top) and head (forward) speed (bottom) for all 10 conditions. Head-LR and Head-UD are head rotated left-right and up-down respectively while walking. B: Top: Functional Gait assessment scores for all subjects, ordered in ascending age (65, 65, 67,71,79); data for mean aged subject groups adapted Walker 2007 for the ages 70-79 (squares) and 80-90 (triangles). Bottom: Maximum lateral step placement for all 6 FGA trials that depend on lateral step. The horizontal dotted lines indicate the limit values for FGA scoring (see the text). C: Proposed additional measures of maximum, and mean step width and movement duration (top, middle and bottom), applicable to all trials along with limits for scoring shown in horizontal dotted lines.

## Discussion

This study demonstrated the feasibility of a low-cost, AI-driven motion capture system to perform functional gait assessment. Through experiments on young and older adults, we showed that this system could accurately capture step placements and achieved a level of accuracy for step width and step length comparable to motion capture. Furthermore, the system's utility was tested through Functional Gait Assessment (FGA) and provided suitable data comparable to previous data [27]. The total cost of the three added consumer-grade web cameras and USB-extension cables was less than $200 CAD, and the analysis computer was inexpensive, meaning an existing computer at a clinic should be sufficient. Thus, modern open-source software appears ready to enable step-length, width and gait speed clinical testing for between 10% and 1% the cost of standard laboratory-based systems.

A critical question of markerless motion-capture systems is whether they are accurate enough for a given real-world application. The available open-source landmark "keypoint detection" inference models vary in 1) outputs: the set of points that are returned; 2) inputs the training set used to generate the model, in this case, human movements; and 3) details of model implementation, such as model architecture, training, and hyperparameters. Here we show that a system leveraging Google's Mediapipe keypoint detection algorithm had sufficient foot position accuracy during stance phases to produce adequate step lengths and widths.

From a practical deployment perspective, the proposed system closely follows existing FGA procedures. The required 6 m walking distance, task sequence, and total assessment time are comparable to standard FGA administration, with the primary difference being camera-based data capture rather than manual observation. Importantly, automated extraction of step placement and gait speed parameters replaces clinician stopwatch timing and subjective visual judgments such as lateral deviation of steps, potentially reducing inter-rater variability without increasing test duration, at the cost of camera calibration.

Velocity information from head speed was also sufficient for walking tasks. Moreover, we expect the quality of available, open-source inference models to only improve, as new models continue to be developed worldwide [28,29]. Thus, the potential for applications to health and industry will continue to broaden.

Functional Gait Assessment relies heavily on human visual identification of lateral foot placement (Fig 5) and we propose step width as an alternative metric. Step width reflects the difference in lateral position of consecutive steps. Step width magnitude is thought to reflect control of lateral stability [14–18,30], given the instability of human gait in the frontal plane [31]. Step length varies with walking speed [32,33] and gait speed decreases with age [21]. Continuous monitoring of both frontal and lateral steps may be particularly useful during unsteady walking conditions like accelerating, decelerating, or changing directions. Other stride-based techniques such as inertial measurement units are incapable of measuring step length and width and thus have limited use for automatic scoring. Here we propose to use step width in addition to or in place of mediolateral foot placement (Fig 5C). It is widely used in the existing scientific literature [15–18,30], applies to non-straight walking paths, and thus is a more flexible measure of balance control.

Our system also yields simple automated assessments of gait speed. Gait speed is a crucial metric in fall risk assessment and traditional approaches often involve only average speed measurements over set distances, relying on manual timing by clinician. This process is repeated multiple times to observe deviations or deterioration (if any) in gait speed. Furthermore, gait speed is combined with other measures such as trunk or head acceleration variability or stride length variability and used to predict fall risk. Our system allows for a more detailed analysis by capturing speed trajectories over varying distances and conditions, such as those in FGA tests (particularly change in gait speed condition, Fig 5) as opposed to single speed measurement. Metrics like peak speed, mean speed, acceleration, and deceleration durations could be quantified from the trajectories. Therefore, our system can offer new metrics of speed for assessing gait speed changes and fall risk. Furthermore, these metrics can be compared with optimal control models [23,34,35] that predict speed and may provide testable mechanistic explanations underlying the difference between older and young adults.

FGA is a comprehensive test, containing conditions involving turns and stepping over obstacles. Our system detects 3D position of body keypoints via 3 cameras that span the walking space, we anticipate no issues with detecting the body throughout 180° turns, and observed no issues in either conditions 4 or 10 of the FGA which include them. Given our system is capable of capturing these movements, we presume that it can effectively be used for other tests such as Timed Up and Go (only visually monitored with a stopwatch) and The Edinburgh Visual Gait score which also require an attentive clinician and may need substantial processing time [3], and still may lack of inter-rater reliability [36,37]. Future studies may perform a systematic assessment of markerless gait analysis on such clinical tests.

One limitation of our study's applicability is the need for camera calibration, and future studies may seek to simplify it or provide streamlined hardware that eliminates it entirely. The current system takes some time and training to perform: a single operator—having no required training in computer vision or biomechanics—moves the Charuco board across the entirety of the walking space, giving view of the board to each pair of cameras across the space, for approximately 1–2 minutes total. The Freemocap program automatically estimates camera parameters (focal length, distortion, position and orientation) from this recording, which takes between 5 and 10 minutes in duration on our hardware. If calibration fails (the computational algorithm can fail to converge on camera parameters) or if errors are large (mean pixel errors are reported), the calibration must be repeated. The software provides the number of detected calibration boards and pixel reprojection errors, which in practice provide useful feedback about calibration accuracy. In our experience, operators became proficient after one or two attempts. Yet a system with minimal impact on the clinician's time and attention would consume no time. Future studies may seek to simplify calibration and hardware setup. One possibility for future software is to provide feedback specific to the clinical task: calibration may be sufficient for the purpose of a clinical gait assessment when random errors are in the range of centimeters, and distances are unbiased. Additionally, practical strategies could mitigate calibration burden down to a single session. If a clinical setup permits cameras to be installed as permanent fixtures— cameras can remain completely undisturbed—then calibration need only be completed a single time. While calibration introduces one additional setup step, similar preparation is standard for many instrumented clinical tools, and the time requirement here is substantially lower than that of laboratory-based motion capture systems.

This study is also limited by sample size. The scope of this project was to use automated gait analysis of markerless data to systematize functional gait assessments (FGA) with accurate foot placement and gait speed. The FGA test is comprehensive, with multiple dependent measures across 10 conditions. Therefore, statistically significant differences across individuals and conditions, with appropriate control for type-1 error rates was beyond the scope of this study. The present results therefore serve primarily as proof of principle that clinically relevant gait metrics can be automatically extracted with limited clinician intervention. Future work involving larger cohorts and patient populations will be required to determine whether these automated measures reliably track the changes in gait behavior over time to predict fall risk and support interventions.

The degree to which clothing affects the performance of markerless motion capture systems remains an open question and likely depends on the dependent measures. A recent study suggests that clothing has a negligible impact on clinically meaningful interpretations [38], but the range of clothing variance was limited to 2 different conditions; "sport" condition (gym shirt + shorts) and "street" condition (unrestricted casual clothing) but did not test baggy clothes, nor the range of clothing variability across different cultures. This study was performed in a controlled laboratory setting in young healthy adults, limiting generalizability to more diverse clothing types and environments. Others report that low visual contrast between the participant and the walking surface particularly at the feet can impair tracking accuracy [39]. In our experience, using shoes that contrast with the ground color and clothing that contrasts with the background and fits relatively tightly to the body can substantially improve tracking performance. For clinical applications, we would recommend performing a brief pilot trial prior to data collection, after which participants can be provided with contrasting garments if necessary to ensure reliable tracking, and a systematic approach to clothing affects with respect to gait speed and step placement is one potential future extension of this study.

One potential future use for markerless motion capture is to integrate inertial measurement units (IMUs). Combining data from IMUs mounted on the feet (measuring linear acceleration and rotational velocity of the feet) can enhance the motion capture accuracy. Under normal conditions, movement can be measured with IMUs [23,35], but absolute orientation and location errors from accelerometer drift enlarge quickly in time without registration. Much larger capture volumes and perhaps multiple simultaneous subject recordings may also be realized with a combined setup. For instance, assuming two cameras are sufficient to record an area of 2 meters by 2 meters, 10 cameras might be integrated to cover a large real-world coordinate system and provide body position data across a large functional region. This setup may significantly improve the usage of IMUs and can facilitate various experiments to investigate how individuals navigate and control their step length, width, and speed in larger environments.

This study validated the feasibility of a low-cost motion capture system powered by computer vision and web cameras. With continued development, this system has the potential to transform clinical gait assessments by providing precise, automated measurements of key metrics such as step width and speed. It offers a cost-effective solution to reduce clinician workload, reduce measurement variability, and make advanced gait analysis accessible to a broader range of healthcare settings.

## Author contributions

**Conceptualization:** Osman Darici, Jeremy D. Wong.

**Data curation:** Chanel Cabak, Jeremy D. Wong.

**Formal analysis:** Chanel Cabak, Jeremy D. Wong.

**Funding acquisition:** Jeremy D. Wong.

**Investigation:** Osman Darici, Chanel Cabak, Jeremy D. Wong.

**Methodology:** Osman Darici, Jeremy D. Wong.

**Project administration:** Chanel Cabak, Jeremy D. Wong.

**Resources:** Jeremy D. Wong.

**Software:** Osman Darici, Jeremy D. Wong.

**Supervision:** Osman Darici, Jeremy D. Wong.

**Validation:** Jeremy D. Wong.

**Visualization:** Jeremy D. Wong.

**Writing – original draft:** Osman Darici, Jeremy D. Wong.

**Writing – review & editing:** Osman Darici, Chanel Cabak, Jeremy D. Wong.

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
