## [Decision Letter · Decision Letter 0]

29 Dec 2025

PONE-D-25-59315A Low-Cost Markerless motion capture system to automate Functional Gait Assessment: Feasibility StudyPLOS One

Dear Dr. Wong,

Thank you for submitting your manuscript to PLOS ONE. After careful consideration, we feel that it has merit but does not fully meet PLOS ONE’s publication criteria as it currently stands. Therefore, we invite you to submit a revised version of the manuscript that addresses the points raised during the review process. Please submit your revised manuscript by Feb 12 2026 11:59PM. If you will need more time than this to complete your revisions, please reply to this message or contact the journal office at plosone@plos.org. Please include the following items when submitting your revised manuscript:

We look forward to receiving your revised manuscript.

Kind regards,

Monika Błaszczyszyn

Academic Editor

PLOS One

Journal Requirements:

2. Please note that PLOS One has specific guidelines on code sharing for submissions in which author-generated code underpins the findings in the manuscript. In these cases, all author-generated code must be made available without restrictions upon publication of the work. Please review our guidelines at https://journals.plos.org/plosone/s/materials-and-software-sharing#loc-sharing-code and ensure that your code is shared in a way that follows best practice and facilitates reproducibility and reuse.

“NSERC Discovery

University of Calgary Catalyst Grant

University of Calgary Biomechanics Research Summer Scholarship”

8. We note that Figure 1 includes an image of a participant in the study.

Reviewers' comments:

Reviewer's Responses to Questions

**Comments to the Author**

1. Is the manuscript technically sound, and do the data support the conclusions?

Reviewer #1: Partly

Reviewer #2: Partly

2. Has the statistical analysis been performed appropriately and rigorously? 

Reviewer #1: No

Reviewer #2: No

3. Have the authors made all data underlying the findings in their manuscript fully available?

Reviewer #1: Yes

Reviewer #2: No

4. Is the manuscript presented in an intelligible fashion and written in standard English?

Reviewer #1: Yes

Reviewer #2: Yes

5. Review Comments to the Author

Reviewer #1: 1-Clarify the novelty of the study

Strengthen the Introduction by explicitly distinguishing how your system meaningfully advances existing open-source or low-cost gait analysis tools.

Suggested location: Introduction, paragraph starting at line 64.

2-Improve the justification for the sample sizes

Explain why N=3 (validation) and N=5 (FGA experiment) are sufficient for a feasibility study and discuss limitations more explicitly.

Suggested location: Methods – Experiment 1 and Experiment 2.

3-Provide clearer participant characteristics

Add demographic details (sex, height, weight, functional ability) to improve transparency and reproducibility.

Suggested location: Methods, lines 167–168.

4-Enhance the description of the experimental validation

Provide a clearer explanation of how synchronization between the marker-based and markerless systems was ensured.

Suggested location: Lines 157–160.

5-Clarify limitations related to calibration accuracy

Expand on how calibration errors might influence step length, width, and speed measurements in real-world applications.

Suggested location: Discussion, lines 309–315.

6-Report uncertainty metrics

Include error margins, RMSE, or Bland–Altman plots in addition to R² to better quantify agreement between systems.

Suggested location: Results, Experiment 1.

7-Strengthen statistical rigor

Even in a feasibility study, add basic statistical comparisons or confidence intervals to reinforce claims of accuracy and consistency.

Suggested location: Lines 223–236.

8-Detail the automated FGA scoring algorithm

Provide more transparency on threshold choices, decision boundaries, and any empirical justification.

Suggested location: Lines 189–220.

9-Clarify how step width and step placement were computed

Present the exact mathematical definitions or formulas used for step width and step length extraction.

Suggested location: Data analysis section (lines 180–187).

10-Improve figure readability

Increase resolution, add clearer legends, and use consistent color coding for left/right steps and conditions.

Suggested location: Figures 1–4.

11-Expand the discussion on clinical applicability

Discuss practical deployment issues: required space, operator training, setup time, and how it compares with existing clinical workflows.

Suggested location: Discussion, lines 269–278.

12-Include reproducibility information

Provide direct links to the repository, software version, dependencies, and detailed instructions for replicating the setup.

Suggested location: Methods or separate Data Availability section.

13-Clarify the ability to detect turning and obstacle tasks

Provide a more explicit explanation of how the system handles 3D orientation changes during turning and obstacle negotiation.

Suggested location: Results, Figure 4 and related text.

14-Improve readability by shortening overly long paragraphs

Break long blocks of text (especially in the Introduction and Discussion) into smaller, thematic paragraphs.

15-Add future work directions more explicitly

Highlight next steps, including larger cohorts, integration with IMUs, and potential clinical trials.

Suggested location: end of Discussion

Reviewer #2: Dear Authors

Thank you for submitting your work to this journal. This manuscript presents a feasibility study on using a low-cost, markerless motion capture system based on open-source tools (FreeMoCap) to automate Functional Gait Assessment (FGA). The authors compare the system against a traditional marker-based motion capture setup and demonstrate its ability to extract key gait parameters (step length, width, and speed) with high accuracy. They also apply the system to older adults performing FGA, showing promising results and proposing supplementary metrics for clinical use. The work is timely, addresses a relevant clinical need for accessible and objective gait assessment tools, and is well-supported by data from small but appropriate sample groups. I think this work is a valuable study and has many novel aspects. Although I believe my comments can improve your manuscript.

Strengths and valuable points:

The study leverages modern, open-source computer vision tools to address a real-world clinical problem to find the best way to analyze.

Validation against a gold-standard marker-based system is methodologically sound and shows strong correlations (R^2 > 0.95).

The proposed automated scoring system for FGA is a logical extension and could reduce subjectivity in clinical assessments.

The cost-effectiveness, comfort, and accessibility of the system are clearly highlighted, making it feasible for wider adoption.

The discussion of supplementary metrics (e.g., step width, continuous speed trajectories) adds value beyond mere automation and gives biomechanical information.

Major Comments:

1- Sample Size Justification:

The study uses very small sample sizes (N=3 for validation, N=5 for FGA). While this is acceptable for a feasibility study, the authors should explicitly acknowledge this limitation and recommend larger-scale validation in future work. Statistical power and generalizability cannot be inferred from such small groups. Although I know preparing and bringing these types of subjects is very difficult. Therefore, you can mention the difficulty of this process in your manuscript.

2- Technical and Practical Limitations:

The calibration process, while described, may still be a barrier for clinical adoption. The authors should discuss potential simplifications or alternatives for routine clinical use.

The sensitivity to clothing, lighting, and shoe contrast is a practical limitation. Suggestions for standardizing test conditions in clinical environments would strengthen the applicability. This information is very important.

Minor Comments:

The manuscript would benefit from a clearer description of how step width is calculated (the difference in mediolateral foot position between consecutive steps) early in the methods. Although, majority of readers may know.

Recommendation:

Minor Revision. The study presents a promising and innovative approach to automating gait assessment. However, the manuscript requires clarification in methods before it can be considered for publication.

Overall, this is a well-conceived and clearly motivated study that aligns with the growing interest in affordable, AI-driven clinical tools. With the above revisions, it will make a valuable contribution to the field. I hope your work will be the starting point of a significant project in the future.

Thank you again for your submissions to this journal

6. PLOS authors have the option to publish the peer review history of their article (what does this mean?). If published, this will include your full peer review and any attached files.

Reviewer #1: No

Reviewer #2: No

---

## [Author Response · Author response to Decision Letter 1]

17 Feb 2026

To the Editor and Reviewers,

We would like to thank the reviewers for their valuable comments. We would in particular like to highlight that both reviewers have identified our lack of clarity with respect to the methods this study, which we have addressed with an additional figure that helps illustrate the processing steps involved in gait extraction and step length and width measurements. We have updated our manuscript substantially based on these comments. Please find our responses to the comments below (in red in the docx file).

Please note: for the reviewer's convenience we inserted low-res figures into the manuscript file, but they continue to be pixelated despite changing Word's DOCX settings. The vector-graphics versions follow the manuscript document.

Reviewer 1

Reviewer #1: 1-Clarify the novelty of the study

Strengthen the Introduction by explicitly distinguishing how your system meaningfully advances existing open-source or low-cost gait analysis tools.

Suggested location: Introduction, paragraph starting at line 64.

We have rewritten the paragraph substantially to better emphasize value in functional gait assessment standardization (paragraph beginning on line 57), as well as the primary gait measures we addressed in this study (step length and width), beginning on line 69.

2-Improve the justification for the sample sizes

Explain why N=3 (validation) and N=5 (FGA experiment) are sufficient for a feasibility study and discuss limitations more explicitly.

Suggested location: Methods – Experiment 1 and Experiment 2.

We acknowledged the low number of subjects as a limitation, discussed the scope of this study and explained our rationale in the paragraph in the Discussion beginning with line 394.

3-Provide clearer participant characteristics

Add demographic details (sex, height, weight, functional ability) to improve transparency and reproducibility.

Suggested location: Methods, lines 167–168.

We have added a table involving the participant information (line 185).

4-Enhance the description of the experimental validation

Provide a clearer explanation of how synchronization between the marker-based and markerless systems was ensured.

Suggested location: Lines 157–164.

We have added a new figure (the new figure 2) about synchronization and provide explanations (lines 157-164 and 201-205).

5-Clarify limitations related to calibration accuracy

Expand on how calibration errors might influence step length, width, and speed measurements in real-world applications.

Suggested location: Discussion, lines 309–315.

In the discussion section we provided information about camera calibration and accuracy (Lines 373-393).

6-Report uncertainty metrics

Include error margins, RMSE, or Bland–Altman plots in addition to R² to better quantify agreement between systems.

Suggested location: Results, Experiment 1.

We updated the figure 3 and included Brand-Altman plots and provided R² and confidence intervals about the measurements (Lines 258 - 274).

7-Strengthen statistical rigor

Even in a feasibility study, add basic statistical comparisons or confidence intervals to reinforce claims of accuracy and consistency.

Suggested location: Lines 223–236.

We reported Brand-Altman plots and provided R² and confidence intervals about the measurements (Lines 258 - 274).

8-Detail the automated FGA scoring algorithm

Provide more transparency on threshold choices, decision boundaries, and any empirical justification.

Suggested location: Lines 189–220.

We provide more clear explanations about the FGA scoring and threshold choices and referred to some papers in the literature (lines 207-232).

9-Clarify how step width and step placement were computed

Present the exact mathematical definitions or formulas used for step width and step length extraction.

Suggested location: Data analysis section (lines 180–187).

We added a new figure (the new figure 2) and illustrated how several parameters including the step width and step length are calculated and provided explanation in the text and figure caption (Lines 165-180).

10-Improve figure readability

Increase resolution, add clearer legends, and use consistent color coding for left/right steps and conditions.

Suggested location: Figures 1–4.

We provided high resolution illustrations of each figure as a .pdf as separate documents within the submission.

11-Expand the discussion on clinical applicability

Discuss practical deployment issues: required space, operator training, setup time, and how it compares with existing clinical workflows.

Suggested location: Discussion, lines 269–278.

In the discussion session, the lines 326-332 and 373-393, we discussed deployment and requirements for the system setup.

12-Include reproducibility information

Provide direct links to the repository, software version, dependencies, and detailed instructions for replicating the setup.

Suggested location: Methods or separate Data Availability section.

In lines 84 to 87, we provided the link to Freemocap software github repository, as well as our data processing pipeline github where code to reproduce the analysis will be made available.

13-Clarify the ability to detect turning and obstacle tasks

Provide a more explicit explanation of how the system handles 3D orientation changes during turning and obstacle negotiation.

Suggested location: Results, Figure 4 and related text.

We provided information about turning, key point detection and potential applications in the lines 364-372 in the discussion section.

14-Improve readability by shortening overly long paragraphs

Break long blocks of text (especially in the Introduction and Discussion) into smaller, thematic paragraphs.

We separated some long paragraphs in the introduction and discussion section.

15-Add future work directions more explicitly

Highlight next steps, including larger cohorts, integration with IMUs, and potential clinical trials.

Suggested location: end of Discussion

In lines 337 to 363 we discussed how the several parameters obtained from our system improve current benefits of FGA. In the lines 364-372, we discussed the applicability of the system to other clinical tests. In the lines 394-402, we mentioned the need to test larger cohorts and finally in the lines 418-428, we discussed the applications with inertial measurement units.

Reviewer 2

Major Comments:

1- Sample Size Justification:

The study uses very small sample sizes (N=3 for validation, N=5 for FGA). While this is acceptable for a feasibility study, the authors should explicitly acknowledge this limitation and recommend larger-scale validation in future work. Statistical power and generalizability cannot be inferred from such small groups. Although I know preparing and bringing these types of subjects is very difficult. Therefore, you can mention the difficulty of this process in your manuscript.

We acknowledged the low number of subjects, discussed its limitations and explained our rationale in the discussion paragraph begins with line 394.

2- Technical and Practical Limitations:

The calibration process, while described, may still be a barrier for clinical adoption. The authors should discuss potential simplifications or alternatives for routine clinical use.

The sensitivity to clothing, lighting, and shoe contrast is a practical limitation. Suggestions for standardizing test conditions in clinical environments would strengthen the applicability. This information is very important.

In the discussion section we provided information about camera calibration and accuracy (Lines 373-393). In the lines 403-417, we provided some information about clothing and markerless motion capture systems from the literature and gave own suggestions.

Minor Comments:

The manuscript would benefit from a clearer description of how step width is calculated (the difference in mediolateral foot position between consecutive steps) early in the methods. Although, majority of readers may know.

We added a new figure (currently Figure 2) and illustrated how several parameters including the step width and step length are calculated and provided explanation in the text and figure caption (Lines 165-180).

Recommendation:

Minor Revision. The study presents a promising and innovative approach to automating gait assessment. However, the manuscript requires clarification in methods before it can be considered for publication.

We provide more clear explanations about the FGA scoring and threshold choices and referred to some papers in the literature (lines 207-232).

---

## [Decision Letter · Decision Letter 1]

23 Mar 2026

A Low-Cost Markerless motion capture system to automate Functional Gait Assessment: Feasibility Study

PONE-D-25-59315R1

Dear Dr. Wong,

We’re pleased to inform you that your manuscript has been judged scientifically suitable for publication and will be formally accepted for publication once it meets all outstanding technical requirements.

Kind regards,

Monika Błaszczyszyn

Academic Editor

PLOS One

Additional Editor Comments (optional):

Reviewers' comments:

Reviewer's Responses to Questions

**Comments to the Author**

1. If the authors have adequately addressed your comments raised in a previous round of review and you feel that this manuscript is now acceptable for publication, you may indicate that here to bypass the “Comments to the Author” section, enter your conflict of interest statement in the “Confidential to Editor” section, and submit your "Accept" recommendation.

Reviewer #1: All comments have been addressed

2. Is the manuscript technically sound, and do the data support the conclusions?

Reviewer #1: (No Response)

3. Has the statistical analysis been performed appropriately and rigorously? 

Reviewer #1: (No Response)

4. Have the authors made all data underlying the findings in their manuscript fully available?

Reviewer #1: (No Response)

5. Is the manuscript presented in an intelligible fashion and written in standard English?

Reviewer #1: (No Response)

6. Review Comments to the Author

Reviewer #1: (No Response)

7. PLOS authors have the option to publish the peer review history of their article (what does this mean?). If published, this will include your full peer review and any attached files.

Reviewer #1: No

---

## [Editor Report · Acceptance letter]

PONE-D-25-59315R1

PLOS One

Dear Dr. Wong,

I'm pleased to inform you that your manuscript has been deemed suitable for publication in PLOS One. Congratulations! Your manuscript is now being handed over to our production team.

Kind regards,

on behalf of

Dr. Monika Błaszczyszyn

Academic Editor

PLOS One